# Long-term cardiovascular disease outcomes in non-hospitalized medicare beneficiaries diagnosed with COVID-19: Population-based matched cohort study

**Quanhe Yang⬚\*, Anping Chang, Xin Tong⬚, Sandra L. Jackson, Robert K. Merritt⬚**

Division for Heart Disease and Stroke Prevention, Centers for Disease Control and Prevention (CDC), Atlanta, Georgia, United States of America

\* qay0@cdc.gov

**Data Availability Statement:** The Medicare beneficiaries' data used in this study to generate the findings are not publicly available and the

## Abstract

### Background

SARS-CoV2, the virus that causes coronavirus disease 2019 (COVID-19), can affect multiple human organs structurally and functionally, including the cardiovascular system and brain. Many studies focused on the acute effects of COVID-19 on risk of cardiovascular disease (CVD) and stroke especially among hospitalized patients with limited follow-up time. This study examined long-term mortality, hospitalization, CVD and stroke outcomes after non-hospitalized COVID-19 among Medicare fee-for-service (FFS) beneficiaries in the United States.

### Methods

This retrospective matched cohort study included 944,371 FFS beneficiaries aged $\geq$66 years diagnosed with non-hospitalized COVID-19 from April 1, 2020, to April 30, 2021, and followed-up to May 31, 2022, and 944,371 propensity score matched FFS beneficiaries without COVID-19. Primary outcomes were all-cause mortality, hospitalization, and incidence of 15 CVD and stroke. Because most outcomes violated the proportional hazards assumption, we used restricted cubic splines to model non-proportional hazards in Cox models and presented time-varying hazard ratios (HRs) and Bonferroni corrected 95% confidence intervals (CI).

### Results

The mean age was 75.3 years; 58.0% women and 82.6% non-Hispanic White. The median follow-up was 18.5 months (interquartile range 16.5 to 20.5). COVID-19 showed initial stronger effects on all-cause mortality, hospitalization and 12 incident CVD outcomes with adjusted HRs in 0–3 months ranging from 1.05 (95% CI 1.01–1.09) for mortality to 2.55 (2.26–2.87) for pulmonary embolism. The effects of COVID-19 on outcomes reduced significantly after 3-month follow-up. Risk of mortality, acute myocardial infarction, cardiomyopathy, deep vein thrombosis, and pulmonary embolism returned to baseline after 6-month

authors cannot share these data due to the Data Use Agreement with the Centers for Medicare and Medicaid Services (CMS). Medicare data are available from CMS following a data use request through the third-party vendor, the Research Data Assistance Center (ResDAC). All researchers seeking to obtain the same type of Medicare data should contact ResDAC at http://www.resdac.org for instructions, guidance, and information on the costs associated with accessing CMS data. The authors did not have any special access privileges that other researchers would not have.

**Funding:** The author(s) received no specific funding for this work.

**Competing interests:** NO authors have competing interests.

**Abbreviations:** AFIB, Atrial Fibrillation; AIS, Acute Ischemic Stroke; AMI, Acute Myocardial Infarction; CVD, Cardiovascular disease; CCI, Charlson-Comorbidity Index; CI, confidence interval; DVT, Deep Vein Thrombosis; FFS, fee-for-service; HF, Heart Failure; HR, hazard ratio; IHD, Ischemic Heart Disease; PE, Pulmonary Embolism; PVD, Peripheral Vascular Disease; TIA, Transient Ischemic Attack.

follow-up. Patterns of initial stronger effects of COVID-19 were largely consistent across age groups, sex, and race/ethnicity.

## Conclusions

Our results showed a consistent time-varying effects of COVID-19 on mortality, hospitalization, and incident CVD among non-hospitalized COVID-19 survivors.

## Introduction

Although SARS-CoV2, the virus that causes the coronavirus disease 2019 (COVID-19), primarily affects the lungs, it can damage many other human organs including the cardiovascular and neurologic systems [1–4]. During the early phase of the COVID-19 pandemic, studies focused on the acute effects of COVID-19 on risk of cardiovascular disease (CVD) and stroke especially among hospitalized patients with limited follow-up time [1, 5–7]. More recent studies examined postacute sequelae of SARS-CoV-2 infection or long-term on risk of CVD and stroke among hospitalized or non-hospitalized patients [1, 8–11]. Most studies presented overall (time-averaged) risk during follow-up, whereas few studies examined the potential time-varying effects [12]. and none examined time-varying effects among non-hospitalized patients aged ≥65 years, where most CVD and strokes occur [13].

The results from self-controlled case series studies suggested that effects of COVID-19 on risk of acute myocardial infarction (AMI) and stroke increased significantly within a few weeks following the index date of COVID-19 and appeared to reduce to the baseline afterwards [14, 15]. Since the overwhelming majority of COVID-19 patients have mild symptoms that do not require hospitalization [16], the purpose of this study was to examine the long-term effects of COVID-19 on risk of CVD and stroke among non-hospitalized COVID-19 patients. We generated a retrospective cohort of Medicare fee-for-service (FFS) beneficiaries aged ≥66 years recovering from non-hospitalized COVID-19 and matched controls without COVID-19 from April 1, 2020, to April 30, 2021, and followed-up to May 31, 2022 in the United States. The primary objectives were to examine the long-term and time-varying effects of COVID-19 on risk of CVD and stroke.

## Methods

### Study cohort

**Non-hospitalized medicare FFS beneficiaries diagnosed with COVID-19.** We used the real-time Medicare Geographic Variation (GV) files to identify the FFS beneficiaries diagnosed with non-hospitalized COVID-19. First, among all Medicare beneficiaries enrolled in 2020 and 2021, we selected all FFS beneficiaries with Part A (inpatient claims) and Part B (physician's office claims) enrollments for ≥11 months coverage in 2020 and 2021. Second, we identified all FFS beneficiaries with diagnosed COVID-19 from January 1, 2020, through April 30, 2021 using ICD-10 code U07.1 as either primary or secondary diagnoses in Part A or Part B claims. Third, we merged FFS beneficiaries with diagnosed COVID-19 from Part A and Part B to determine the first date of COVID-19 claims as the starting time of COVID-19 exposure. Fourth, we excluded all FFS beneficiaries who had first diagnosed COVID-19 based on U07.1 before April 1, 2020 for a clean definition of COVID-19 patients (U0.71 was officially started on April 1, 2020). Fifth, for non-hospitalized COVID-19 beneficiaries, we excluded all FFS

beneficiaries who had first diagnosed COVID-19 from Part A only, or the date of COVID-19 in Part A was earlier than the first diagnosis date from Part B claims. Sixth, among the remining FFS beneficiaries, we further excluded beneficiaries who had inpatient claims (Part A) within 14 days of the first date of diagnosed COVID-19 in Part B (probable hospitalized COVID-19). Seventh, we excluded all beneficiaries who died within 28 days of the first date of diagnosed COVID-19, because the primary objective was to examine the long-term effect of COVID-19. Eighth, we excluded all beneficiaries aged <66 years because we need at least 1-year lookback period to determine comorbidity conditions at baseline, and the incident CVD and stroke outcomes during follow up. Ninth, we excluded all beneficiaries who had any inpatient admissions (all-cause) overlapped with the first date of diagnosed COVID-19. Tenth, we excluded beneficiaries with non-hospitalized COVID-19 who didn't match to a control beneficiary after three rounds of rematching. The final cohort consisted of 944,371 FFS beneficiaries with non-hospitalized COVID-19 and 944,371 matched controls (Fig 1).

**FFS beneficiaries without a history of COVID-19.**   To identify beneficiaries without a history of COVID-19 as potential matching controls, first, we excluded all FFS beneficiaries with hospitalized or non-hospitalized COVID-19 as described in the above COVID-19 case selection process. Second, we excluded all FFS beneficiaries with any documentation of ICD-10 code B97.29 (Other coronavirus as the cause of diseases classified

Elsewhere) from January 1 to March 31, 2020 as probable COVID-19 cases, before using U0.71 code from April 1, 2020. Third, we excluded all FFS beneficiaries with any documentation of U0.71 during follow up from April 1, 2020, to May 31, 2022. Fourth, among FFS beneficiaries without a history of COVID-19, we selected all FFS beneficiaries aged 66 years or older who had ≥11 months continuous enrollment in Part A and Part B in 2020 and 2021 or full enrollment before death. Fifth, we excluded FFS beneficiaries with missing information on key matching variables, e.g., age, sex, and race/ethnicity (Fig 1).

**Propensity score matching.**   Propensity score matching was used to select controls who were similar to COVID-19 cases to minimize confounding and selection bias when estimating the long-term effects of COVID-19. We derived the propensity score by using multivariate logistic regression that exactly matched on age, sex, race/ethnicity, long-term stay status, state of residence and included socioeconomic variables, health care utilization characteristics, frailty characteristics, CMS Hierarchical Condition Category score, coexisting health conditions and Charlson-Comorbidity-Index (S1 Table). We matched non-hospitalized COVID-19 beneficiaries to the controls by cohort year (2020 and 2021 cohort, separately). For the 2020 cohort, we used April 1, 2020 as the starting time of lookback for calculating the baseline matching variables, and for the 2021 cohort, we used January 1, 2021. The lookback period was at least one year, depending on the beneficiary's age (e.g., one year for age 66 years, two years for age 67 years and so on). For each matched control beneficiary, we assigned the date of exposure as the same date of the corresponding case beneficiary. If the matched control beneficiaries died within 28 days of the assigned exposure date, we rematched new control beneficiaries up to three times before stopping matching. We used a "greedy nearest neighbor" one-to-one matching on the logit of propensity score with caliper of width 0.2 in SAS Proc PSMATCH procedure (release 9.4; SAS Institute, Cary, NC). The balance of the covariates was assessed by the standardized difference where differences <0.10 were considered negligible [17].

**CVD and stroke outcomes.**   Primary outcomes were all-cause mortality, hospitalization, and incidence of 15 CVD and stroke outcomes including: (1) abnormalities of heart rhythm, (2) acute myocardial infarction (AMI), (3) atrial fibrillation and flutter (AFIB), (4) cardiac arrhythmia, (5) cardiomyopathy, (6) deep vein thrombosis (DVT), (7) pulmonary embolism (PE), (8) heart failure (HF), (9) hypercoagulability, (10) ischemic heart disease (IHD), (11) peripheral vascular disease (PVD), (12) all-stroke, (13) acute ischemic stroke (AIS), (14)

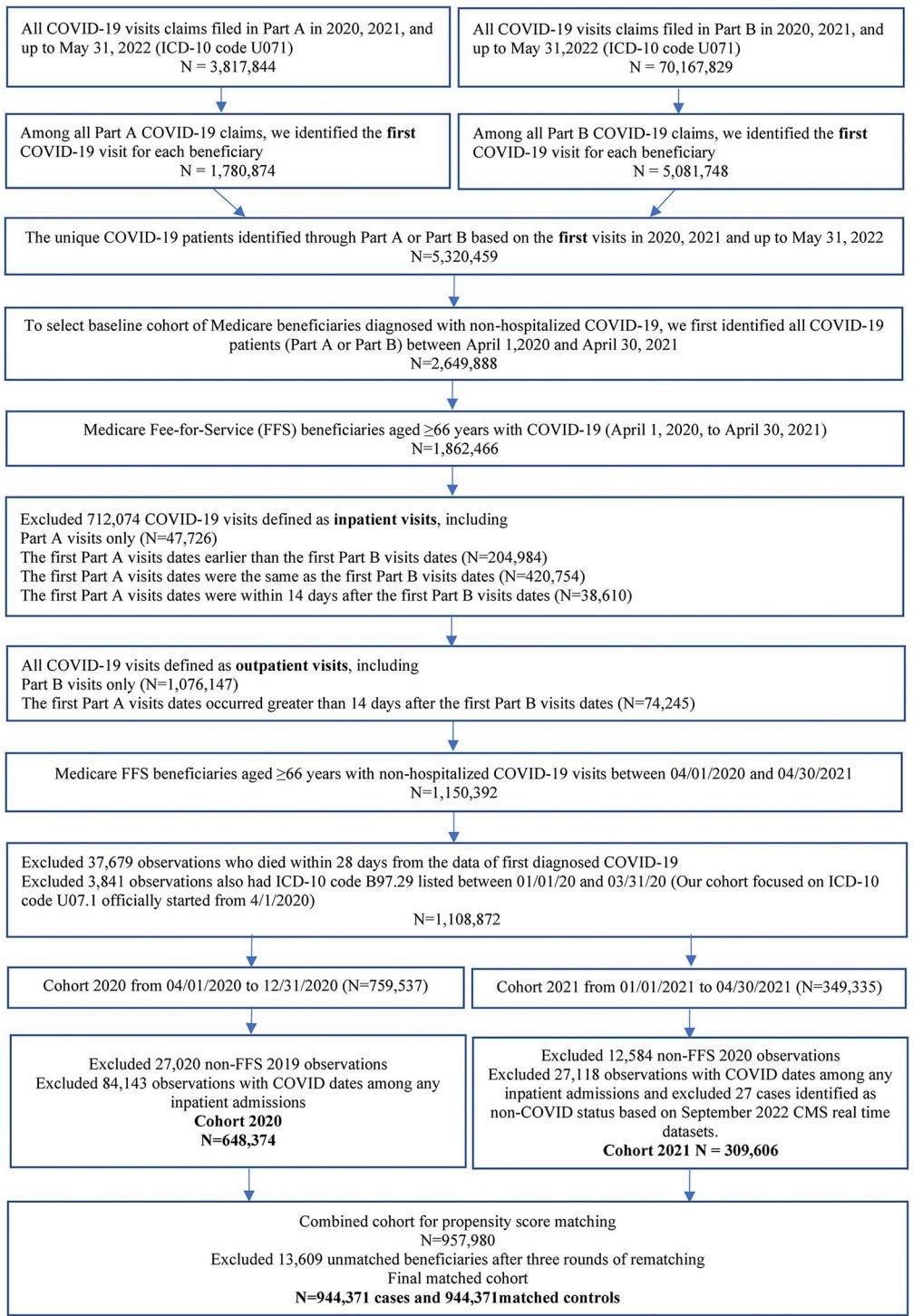

**Fig 1. Flowchart of medicare fee-for-service beneficiaries diagnosed with non-hospitalized COVID-19 and matched controls, Medicare 2020–2021 matched cohort.**

hemorrhagic stroke, and (15) transient ischemic attack (TIA). The secondary outcomes were incidence of any diagnosis of CVD and stroke outcomes among all beneficiaries (including the beneficiaries who had preexisting CVD or stroke at baseline). Where available, we defined

CVD and stroke outcomes based on ICD-10 codes in the Chronic Condition Warehouse (CCW) Condition Algorithms [18]. For conditions that were not included in CCW Condition Algorithms, we used the ICD-10 codes provided in the published literature. Definitions, ICD-10 codes, and detailed criteria to define incident CVD and stroke outcomes during follow-up were documented in S2 Table. The baseline cohort consisted of Medicare FFS beneficiaries with non-hospitalized COVID-19 and matched controls from April 1, 2020 to April 30, 2021, and the end of follow-up was May 31, 2022.

## Statistical analysis

We calculated mean, median and percentages of the selected covariates by COVID-19 status and presented the standardized differences between Medicare FFS beneficiaries with non-hospitalized COVID-19 and matched controls. We calculated average median and interquartile range of follow-up by averaging the median follow-up months across the different outcomes. We used cause-specific Cox models where death was considered as a competing risk to examine the association between the COVID-19 and risk of CVD and stroke [19]. We tested the proportional hazard assumption by using an interaction term of COVID-19 and survival time and by the weighted Schoenfeld residuals plots; 14 of 17 outcomes showed significant non-proportional hazards between COVID-19 and outcomes. We used restricted cubic splines (RCS) with 4-knots of equal number of events to model non-proportional hazards in cause-specific Cox models and presented time-varying hazard ratios (HRs) and Bonferroni corrected 95% confidence intervals (CI) (calculated as $((1-(0.05/N))*100\% = 99.7\%$ CI), where N = 17 was the number of outcomes) to account for multiple comparisons [20].

The RCS Cox models showed a consistent pattern of initial stronger effects of COVID-19 on risk of mortality, hospitalization and CVD during approximately first 3 months of follow-up and reduced effects afterwards. To show the time-varying effects of COVID-19, we estimated the piecewise cause-specific HRs (Bonferroni 95% CIs) and divided the follow-up time into three periods: 0–3, 4–6 and $\geq$7 months from the index date. For the RCS non-proportional and piecewise cause-specific Cox models, we presented fully adjusted HRs that included all matching variables accounting for the potential residual confounding.

For the risk of incident outcomes, we excluded Medicare FFS beneficiaries who had a history of the outcome being examined and calculated the incidence rates per 1,000 person-years. The cohort size varied depending on the incident events. For mortality and hospitalizations, we used full cohort. For secondary analyses, we included beneficiaries who had pre-existing CVD or stroke outcome being examined (defined as incidence of any diagnosis). We conducted stratified analyses by age group (66–74, 75–84 and $\geq$85 years), sex, and race/ethnicity (non-Hispanic White, non-Hispanic Black, Hispanic and Other) and tested for potential effect modification by including interaction terms of COVID-19 by age group, sex, and race/ethnicity in Cox non-proportional hazard models. SAS, version 9.4 (SAS Institute) was used for analysis.

This study used de-identified Medicare claims data that are exempt from IRB review, was reviewed by Centers for Disease Control and Prevention (CDC), and conducted by adhering to applicable federal law, CDC policy, and the CDC-CMS Interagency agreement and data use agreement. This study started accessing the CMS Medicare datasets on July 29, 2021 and completed the study on April 25, 2023.

## Results

### Patient characteristics

The mean age of Medicare FFS beneficiaries was 75.3 years; 58.0% women and 82.6% were non-Hispanic White. The average median follow-up was 18.5 months (interquartile range 16.5

to 20.5) with a maximum of 26 months. Nearly all standardized differences of covariates at baseline between FFS beneficiaries with non-hospitalized COVID-19 and matched controls were less than 0.05 (Table 1) and improved significantly from the matching pool of all Medicare FFS beneficiaries without COVID-19 (S3 Table).

### Incidence rates and risk of CVD and stroke

COVID-19 showed initial stronger effects on all-cause mortality, hospitalization, and incidence of 12 CVD outcomes (except for all-stroke, AIS and hemorrhagic stroke, Fig 2–17). Adjusted HRs in 0–3 months ranged from 1.05 (95% CI 1.01–1.09) for mortality to 2.55 (2.26–2.87) for PE. The corresponding incidence rates per 1,000 person-year were 51.9 (51.0–52.9) among those with COVID-19 vs. 45.2 (44.3–46.1) among those without COVID-19 for mortality and 8.2 (7.9–8.6) vs. 3.1 (2.9–3.4) for PE respectively (Table 2). The initial stronger effects of COVID-19 on outcomes reduced significantly during 4–6 months of follow-up. After ≥7-month follow-up, the adjusted HRs ranged from 1.04 (1.01–1.07) for HF to 1.20 (1.16–1.23) for PVD. The corresponding incidence rates per 1,000 were 22.2 (21.8–22.5) vs. 21.4 (21.0–21.7) for HF and 30.4 (30.0–30.9) vs 25.6 (25.2–26.0) for PVD, respectively. Risk of mortality, AMI, cardiomyopathy, DVT, and PE returned to baseline after 6-month follow-up. (Table 2 and Fig 2–17). There were no significant associations between COVID-19 and risk of all stroke, AIS and hemorrhagic stroke (Table 2).

Patterns of initial stronger effects of COVID-19 were largely consistent across age group, sex and race/ethnicity (S1–S3 Figs). The interactions between age group and mortality, hospitalizations, abnormality of heart rhythm, AFIB, cardiomyopathy, IHD, PVD, and TIA were significant (Bonferroni p<0.05). Older ages were associated with higher risk of mortality and PVD and younger ages appeared to be associated with higher risk of hospitalizations, abnormality of heart rhythm, AFIB, cardiomyopathy, IHD and TIA. There were significant interactions between sex and hospitalization, AMI, HF, and PVD. Men tended to have stronger associations with these outcomes, except for PVD. Interactions between race/ethnicity and hospitalization, abnormal heart rhythms, and cardiomyopathy were significant; non-Hispanic White beneficiaries appeared to have lower risk of these outcomes (Bonferroni p<0.05, S1–S3 Figs).

### Incidence rates of any diagnosis and risk of CVD and stroke

The incidence rates of any diagnosis of CVD and stroke (including the beneficiaries who had pre-existing CVD or stroke at baseline) were higher than that of incidence rates (excluding the beneficiaries with pre-existing CVD or stroke at baseline). However, the patterns of association between COVID-19 and incidence rates of any diagnosis were largely consistent with those of the incident rates excluding the beneficiaries with pre-existing CVD or stroke (S4 Table and S4 Fig).

## Discussion

In this population-based cohort study of over 1.8 million Medicare FFS beneficiaries with non-hospitalized COVID-19 and matched controls during a maximum of 26 months of follow-up, COVID-19 showed a consistent pattern of initial stronger effect (time-varying effects), with the strongest effects on all-cause mortality, hospitalization, and incident CVD occurring within the first 3 months post-COVID-19. The risk of mortality, hospitalization, CVD and TIA reduced significantly after 3-months from the index date of COVID-19, although risk remained elevated for several outcomes. Patterns of initial stronger effects of COVID-19 on the outcomes were largely consistent across age group, sex and race/ethnicity.

**Table 1. Characteristics of Medicare fee-for-service beneficiaries aged 66 years or older with non-hospitalized COVID-19 and matched controls, Medicare 2020–2021 matched cohort.**

| Characteristics | Number of FFS beneficiaries with Non-hospitalized COVID-19 | COVID-19 mean/median/% (95% CI) | Number of matched FFS beneficiaries | Non-COVID-19 mean/median/% (95% CI) | Standardized differences [a] |
|---|---|---|---|---|---|
| **All** | 944,371 | | 944,371 | | |
| **Age, mean** | | 75.28 (75.26–75.29) | | 75.28 (75.26–75.29) | 0.00 |
| **Age group (%)** | | | | | |
| 66–74 years | 524,925 | 55.58 (55.48–55.68) | 524,925 | 55.58 (55.48–55.68) | 0.00 |
| 75–84 years | 296,336 | 31.38 (31.29–31.47) | 296,336 | 31.38 (31.29–31.47) | 0.00 |
| ≥85 years | 123,110 | 13.04 (12.97–13.10) | 123,110 | 13.04 (12.97–13.10) | 0.00 |
| **Sex (%)** | | | | | |
| Men | 396,517 | 41.99 (41.89–42.09) | 396,517 | 41.99 (41.89–42.09) | 0.00 |
| Women | 547,854 | 58.01 (57.91–58.11) | 547,854 | 58.01 (57.91–58.11) | 0.00 |
| **Race/ethnicity** | | | | | |
| Non-Hispanic White | 779,549 | 82.55 (82.47–82.62) | 779,549 | 82.55 (82.47–82.62) | 0.00 |
| Non-Hispanic Black | 56,030 | 5.93 (5.89–5.98) | 56,030 | 5.93 (5.89–5.98) | 0.00 |
| Hispanic | 58,264 | 6.17 (6.12–6.22) | 58,264 | 6.17 (6.12–6.22) | 0.00 |
| Other | 50,528 | 5.35 (5.31–5.40) | 50,528 | 5.35 (5.31–5.40) | 0.00 |
| **Low-income subsidy (%)** | | | | | |
| Yes | 156,178 | 16.54 (16.46–16.61) | 151,828 | 16.08 (16.00–16.15) | 0.0125 |
| No | 788,193 | 83.46 (83.39–83.54) | 792,543 | 83.92 (83.85–84.00) | |
| **Household Income, Median (IQR)** | | 75,566 (60,954–100,909) | | 76,288 (61,370–100,973) | -0.0063 |
| **Social Vulnerability Index, Median (IQR)** | | 0.54 (0.29–0.74) | | 0.53 (0.31–0.73) | 0.0008 |
| **Ischemic Heart Disease (%)** | | | | | |
| Yes | 455,368 | 48.22 (48.12–48.32) | 449,400 | 47.59 (47.49–47.69) | 0.0127 |
| No | 489,003 | 51.78 (51.68–51.88) | 494,971 | 52.41 (52.31–52.51) | |
| **Acute Myocardial Infarction (%)** | | | | | |
| Yes | 43,297 | 4.58 (4.54–4.63) | 40,799 | 4.32 (4.28–4.36) | 0.0128 |
| No | 901,074 | 95.42(95.37–95.46) | 903,572 | 95.68 (95.64–95.72) | |
| **Congestive heart failure (%)** | | | | | |
| Yes | 237,298 | 25.13 (25.04–25.22) | 226,453 | 23.98 (23.89–24.07) | 0.0267 |
| No | 707,073 | 74.87 (74.78–74.96) | 717,918 | 76.02 (75.93–76.11) | |
| **Peripheral Vascular Disease (%)** | | | | | |
| Yes | 242,314 | 25.66 (25.57–25.75) | 232,911 | 24.66 (24.58–24.75) | 0.0229 |
| No | 702,057 | 74.34 (74.25–74.43) | 711,460 | 75.34 (75.25–75.42) | |
| **Atrial fibrillation (%)** | | | | | |
| Yes | 149,813 | 15.86 (15.79–15.94) | 144,560 | 15.31 (15.23–15.38) | 0.0153 |
| No | 794,558 | 84.14 (84.06–84.21) | 799,811 | 84.69 (84.62–84.77) | |
| **Hypertension (%)** | | | | | |
| Yes | 769,036 | 81.43 (81.36–81.51) | 773,812 | 81.94 (81.86–82.02) | -0.0131 |
| No | 175,335 | 18.57 (18.49–18.64) | 170,559 | 18.06 (17.98–18.14) | |
| **Hyperlipidemia (%)** | | | | | |
| Yes | 780,417 | 82.64 (82.56–82.72) | 786,715 | 83.31 (83.23–83.38) | -0.0177 |
| No | 163,954 | 17.36 (17.28–17.44) | 157,656 | 16.69 (16.62–16.77) | |
| **Stroke/TIA (%)** | | | | | |
| Yes | 138,937 | 14.71 (14.64–14.78) | 133,209 | 14.11 (14.04–14.18) | 0.0173 |
| No | 805,434 | 85.29 (85.22–85.36) | 811,162 | 85.89 (85.82–85.96) | |

(*Continued*)

**Table 1.** (Continued)

| Characteristics | Number of FFS beneficiaries with Non-hospitalized COVID-19 | COVID-19 mean/median/% (95% CI) | Number of matched FFS beneficiaries | Non-COVID-19 mean/median/% (95% CI) | Standardized differences [a] |
|---|---|---|---|---|---|
| **Diabetes (%)** | | | | | |
| Yes | 376,471 | 39.86 (39.77–39.96) | 373,032 | 39.50 (39.40–39.60) | 0.0074 |
| No | 567,900 | 60.14 (60.04–60.23) | 571,339 | 60.50 (60.40–60.60) | |
| **COPD (%)** | | | | | |
| Yes | 235,648 | 24.95 (24.87–25.04) | 225,656 | 23.89 (23.81–23.98) | 0.0239 |
| No | 708,723 | 75.05 (74.96–75.13) | 718,715 | 76.11 (76.02–76.19) | |
| **Alzheimer (%)** | | | | | |
| Yes | 70,488 | 7.46 (7.41–7.52) | 60,717 | 6.43 (6.38–6.48) | 0.0407 |
| No | 873,883 | 92.54 (92.48–92.59) | 883,654 | 93.57 (93.52–93.62) | |
| **Obesity (%)** | | | | | |
| Yes | 325,046 | 34.42 (34.32–34.52) | 325,857 | 34.51 (34.41–34.60) | -0.0018 |
| No | 619,325 | 65.58 (65.48–65.68) | 618,514 | 65.49 (65.40–65.59) | |
| **Tobacco use (%)** | | | | | |
| Yes | 96,861 | 10.26 (10.20–10.32) | 92,763 | 9.82 (9.76–9.88) | 0.0144 |
| No | 847,510 | 89.74 (89.68–89.80) | 851,608 | 90.18 (90.12–90.24) | |
| **Charlson Comorbidity Index (%)** | | | | | |
| 0 | 775,040 | 82.07 (81.99–82.15) | 799,939 | 84.71 (84.63–84.78) | -0.0709 |
| 1 | 46,675 | 4.94 (4.90–4.99) | 42,624 | 4.51 (4.47–4.56) | 0.0202 |
| 2 | 36,765 | 3.89 (3.85–3.93) | 32,721 | 3.46 (3.43–3.50) | 0.0227 |
| 3 | 26,008 | 2.75 (2.72–2.79) | 22,305 | 2.36 (2.33–2.39) | 0.0248 |
| 4 | 18,094 | 1.92 (1.89–1.94) | 15,180 | 1.61 (1.58–1.63) | 0.0235 |
| ≥5 | 41,789 | 4.43 (4.38–4.47) | 31,602 | 3.35 (3.31–3.38) | 0.0558 |

Abbreviations: CI, confidence interval; FFS, fee-for-service; TIA, transient ischemic stroke.

[a] Standardized differences were the differences in means or proportions between non-hospitalized COVID-19 FFS beneficiaries and matched FFS beneficiaries divided by standard errors where differences <0.10 were considered negligible.

## Comparison with other studies

Many studies examined long-term CVD outcomes of COVID-19 among hospitalized and non-hospitalized COVID-19 patients with varying length of follow-up period and showed increased risk of CVD and stroke among COVID-19 survivors [1, 3, 5, 6, 8–11, 14]. Our findings were largely consistent with the results of other studies, when considering average HRs across the follow-up period. A recent cohort study by Emma Rezel-Potts, et al. examined long-term effects of COVID-19 on risk of CVD and diabetes using electronic records of 1,356 patients in UK family practices. The study showed time-varying effects of COVID-19 and found that CVD increased within 12 weeks after COVID-19 and returned to baseline from 13 to 52 weeks [12]. A Swedish study used the self-controlled case series (SCCS) design and matched cohort to examine risk of AMI and ischemic stroke following COVID-19 using linked data from the national registers for inpatient and outpatient clinics. The study showed stronger effects of COVID-19 on risk of AMI and stroke within 2 weeks after COVID-19 [14]. Another study using the SCCS design showed significantly higher risk of AIS within 7 days after COVID-19 among Medicare FFS beneficiaries aged ≥65 years [15]. The findings of our study provided further evidence of time-varying effects of COVID-19 on mortality, hospitalization, CVD and TIA among Medicare FFS beneficiaries with non-hospitalized COVID-19. The effects of COVID-19 decreased for 14 of 17 outcomes over time, and became

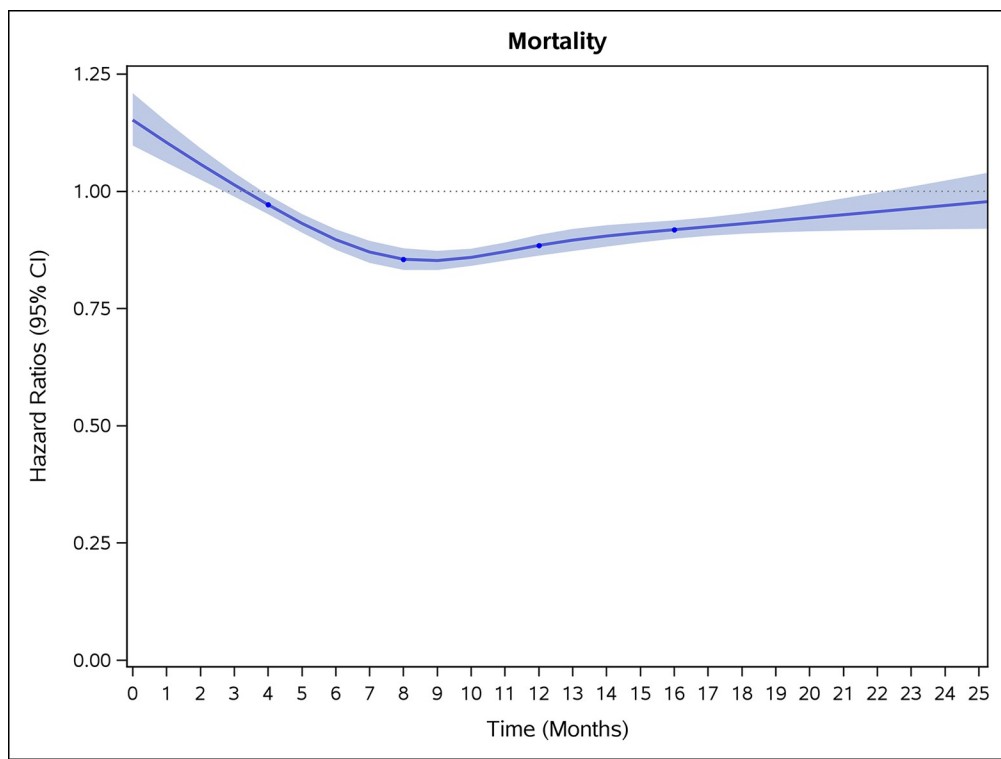

**Fig 2. Time-varying hazard ratios (95% CI) for risk of death associated with non-hospitalized COVID-19, Medicare 2020–2021 matched cohort.**

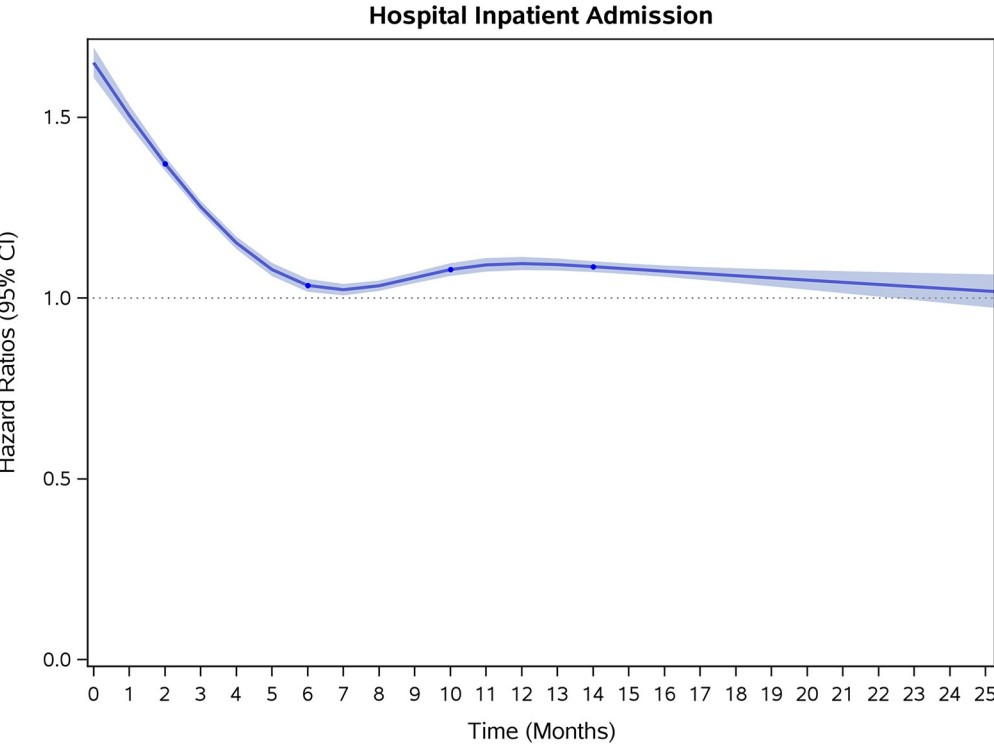

**Fig 3. Time-varying hazard ratios (95% CI) for risk of hospitalization associated with non-hospitalized COVID-19, Medicare 2020–2021 matched cohort.**

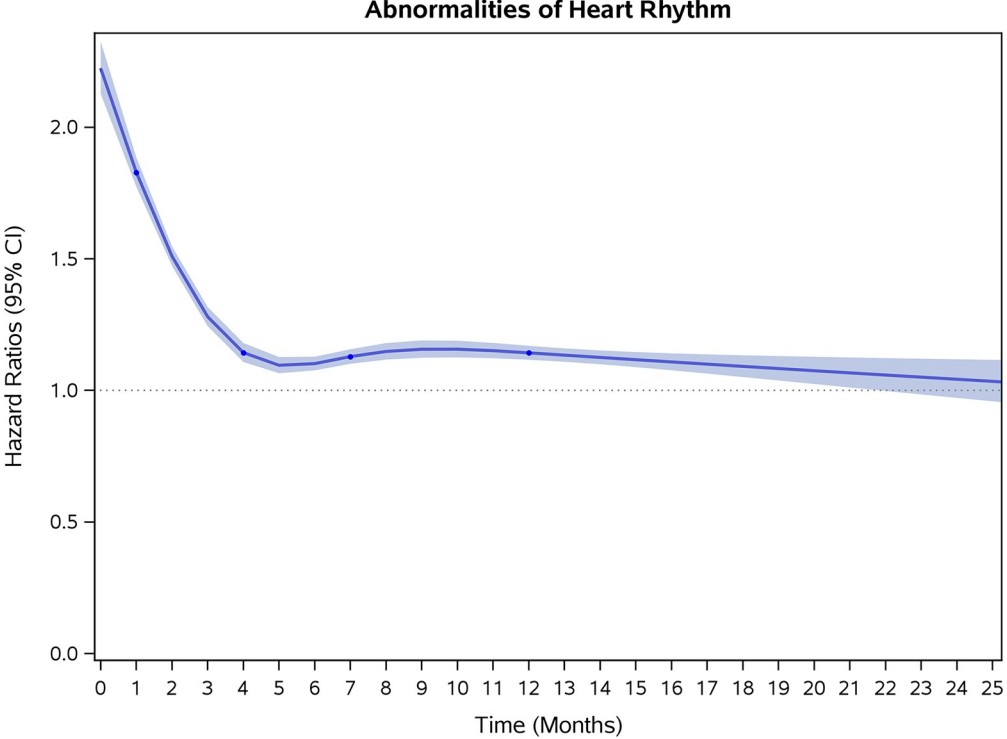

**Fig 4. Time-varying hazard ratios (95% CI) for risk of incident abnormalities of heart rhythm associated with non-hospitalized COVID-19, Medicare 2020–2021 matched cohort.**

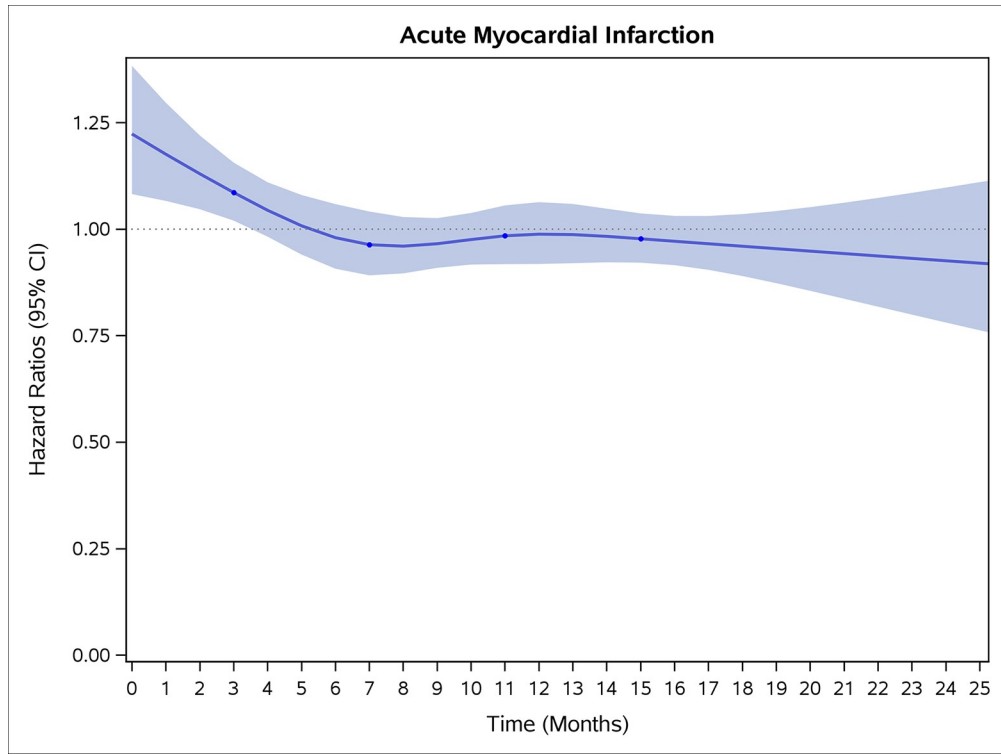

**Fig 5. Time-varying hazard ratios (95% CI) for risk of incident acute myocardial infarction associated with non-hospitalized COVID-19, Medicare 2020–2021 matched cohort.**

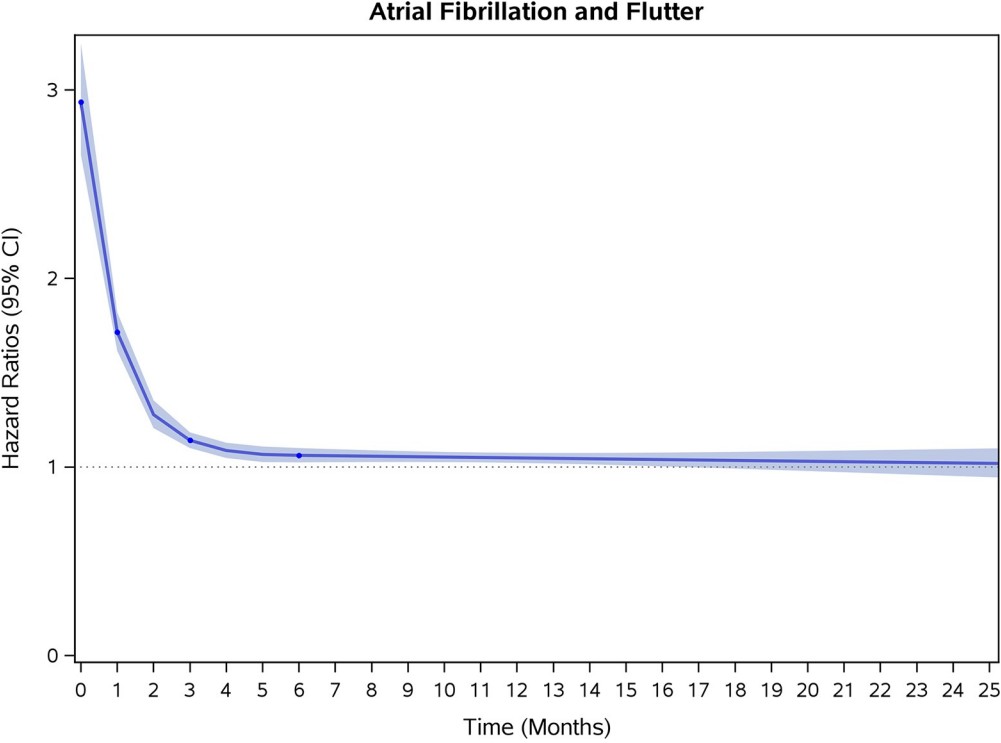

**Fig 6. Time-varying hazard ratios (95% CI) for risk of incident atrial fibrillation and flutter associated with non-hospitalized COVID-19, Medicare 2020–2021 matched cohort.**

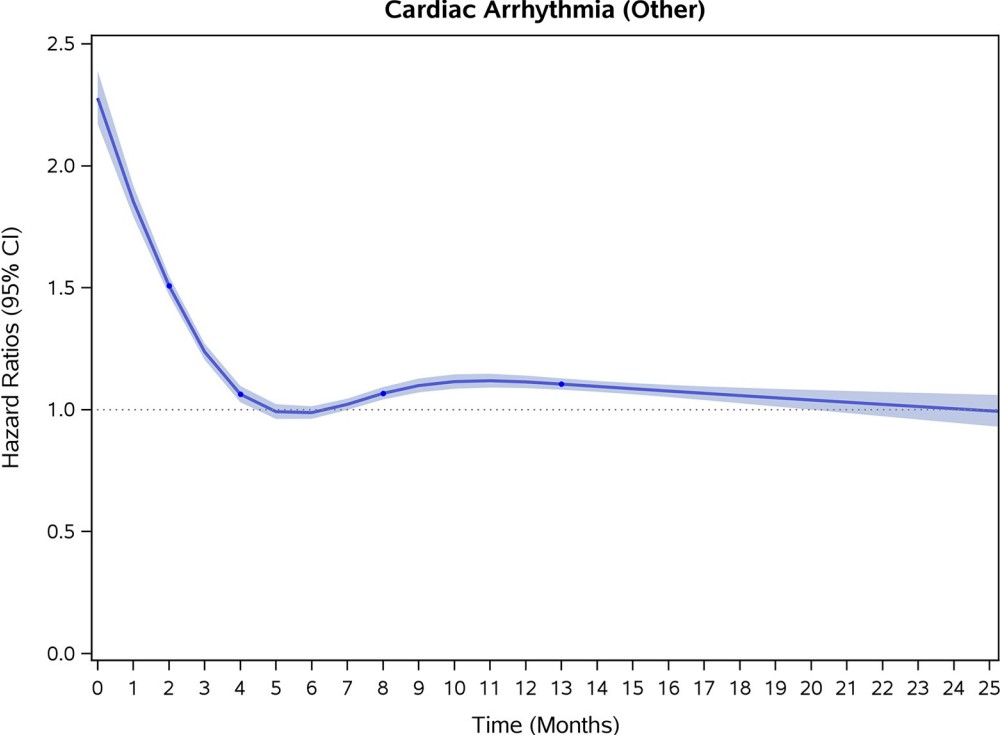

**Fig 7. Time-varying hazard ratios (95% CI) for risk of incident cardiac arrhythmia associated with non-hospitalized COVID-19, Medicare 2020–2021 matched cohort.**

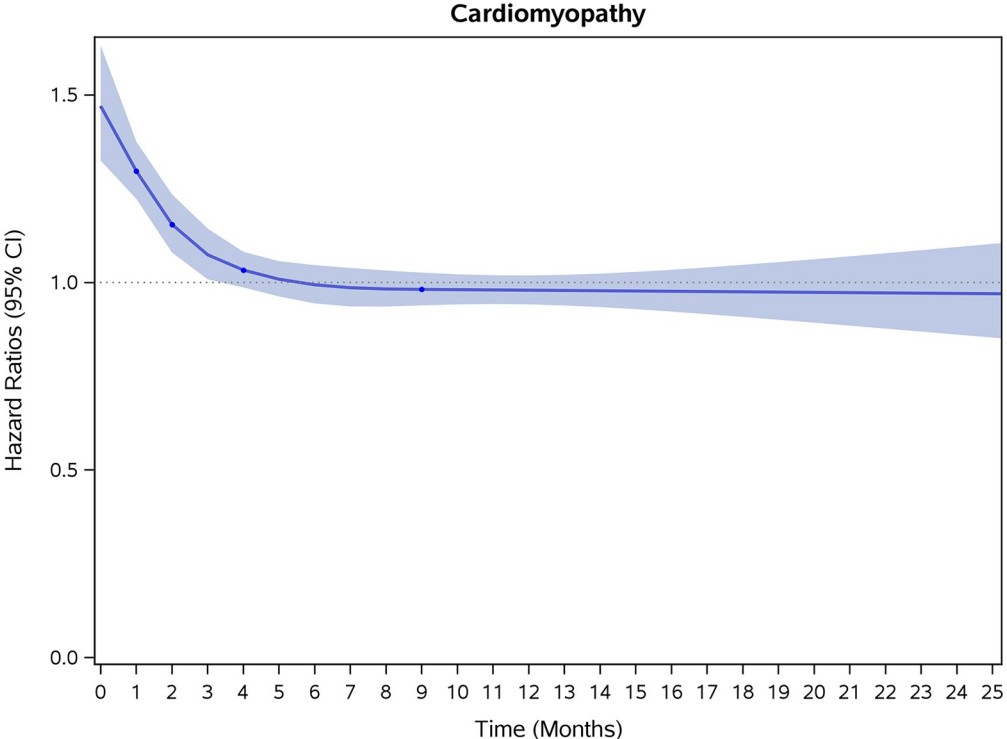

**Fig 8. Time-varying hazard ratios (95% CI) for risk of incident cardiomyopathy associated with non-hospitalized COVID-19, Medicare 2020–2021 matched cohort.**

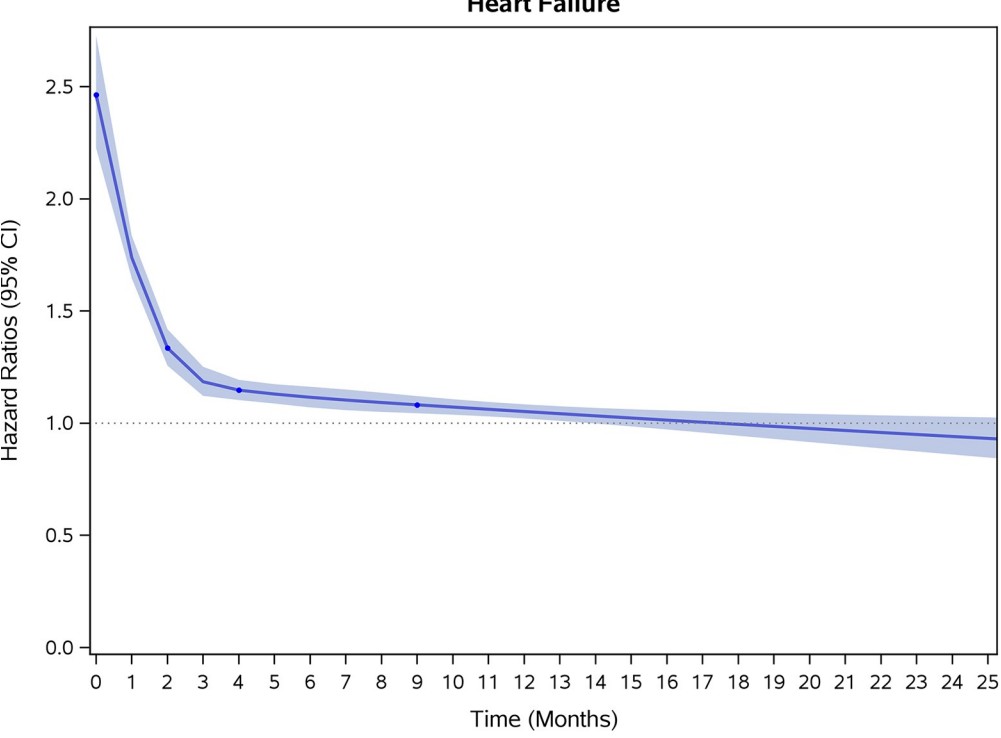

**Fig 9. Time-varying hazard ratios (95% CI) for risk of incident heart failure associated with non-hospitalized COVID-19, Medicare 2020–2021 matched cohort.**

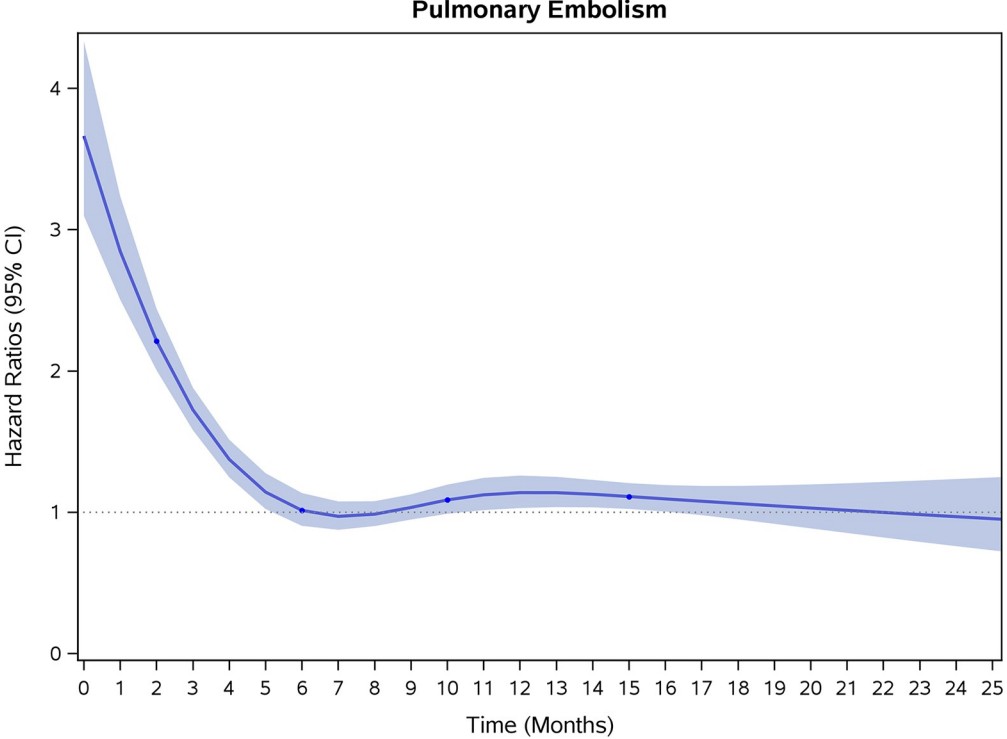

**Fig 10. Time-varying hazard ratios (95% CI) for risk of incident pulmonary embolism associated with non-hospitalized COVID-19, Medicare 2020–2021 matched cohort.**

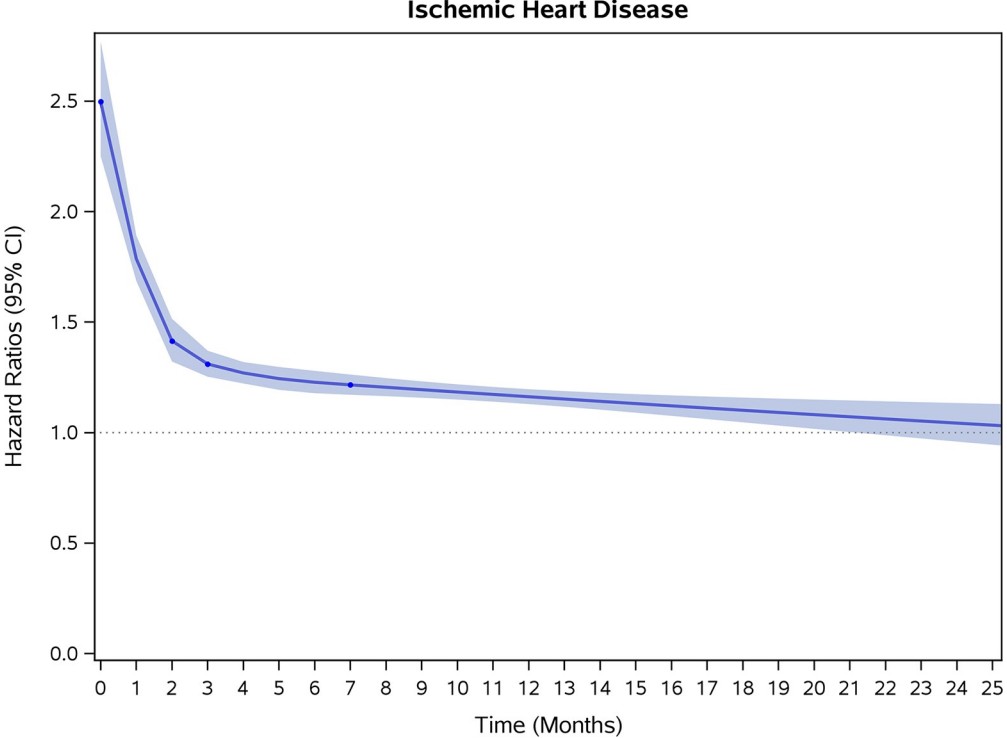

**Fig 11. Time-varying hazard ratios (95% CI) for risk of incident ischemic heart disease associated with non-hospitalized COVID-19, Medicare 2020–2021 matched cohort.**

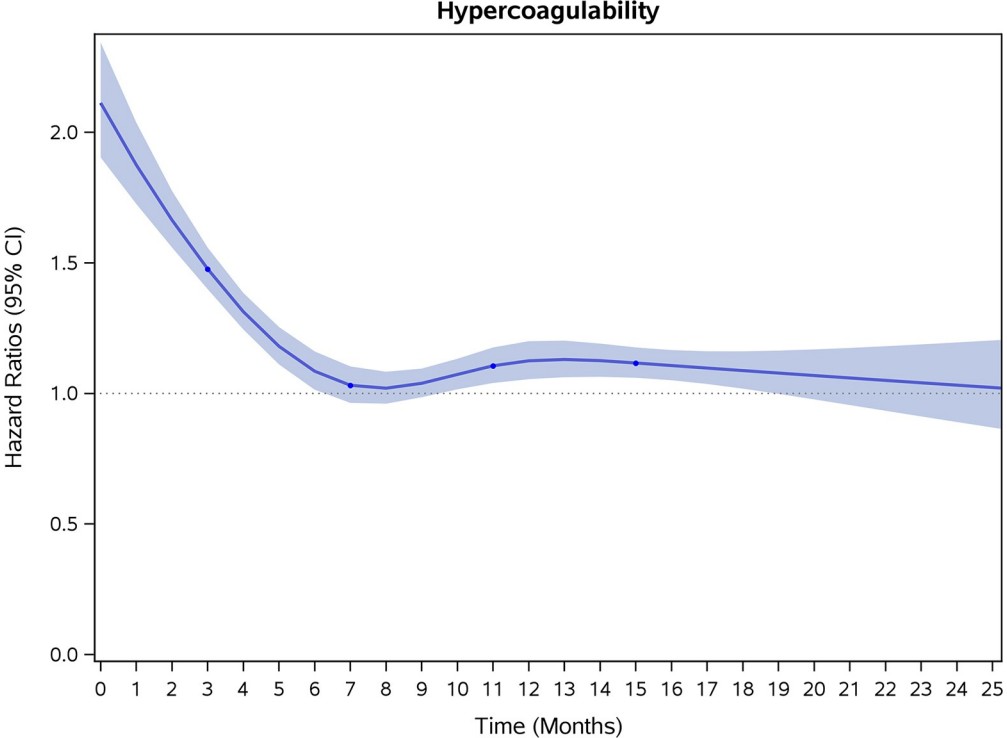

**Fig 12. Time-varying hazard ratios (95% CI) for risk of incident hypercoagulability associated with non-hospitalized COVID-19, Medicare 2020–2021 matched cohort.**

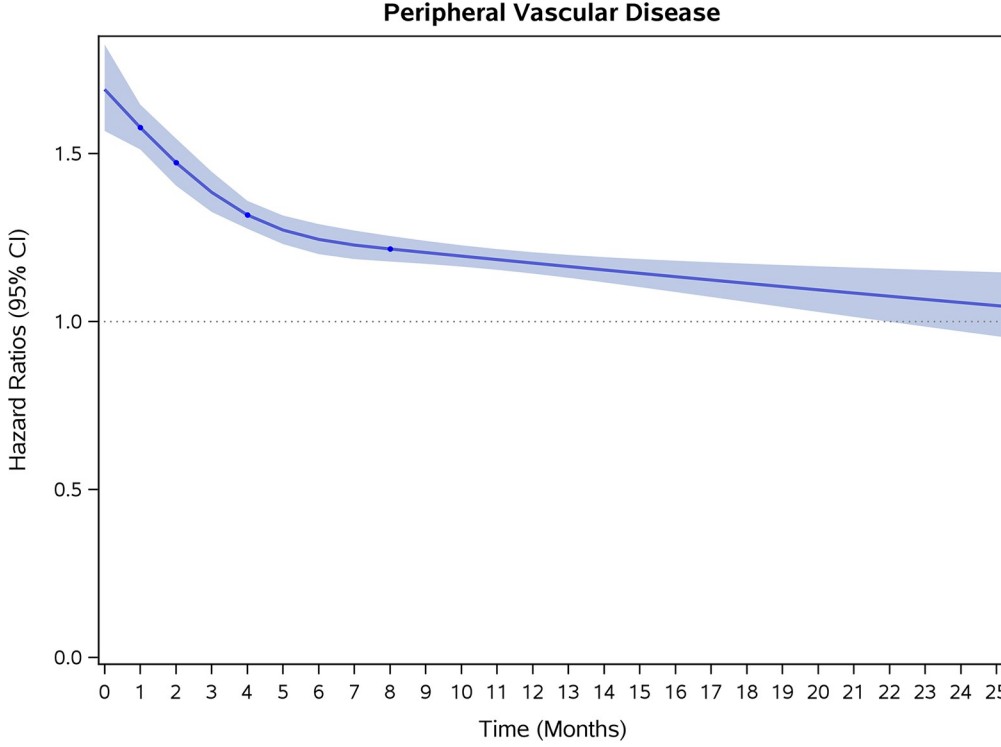

**Fig 13. Time-varying hazard ratios (95% CI) for risk of incident peripheral vascular disease associated with non-hospitalized COVID-19, Medicare 2020–2021 matched cohort.**

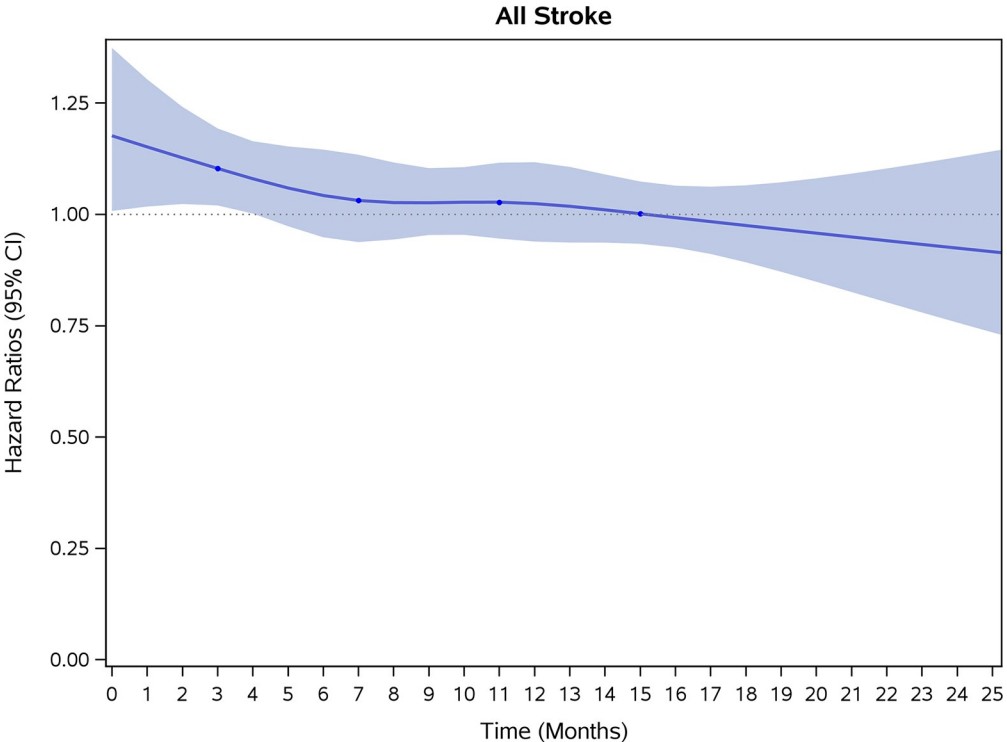

**Fig 14. Time-varying hazard ratios (95% CI) for risk of incident stroke associated with non-hospitalized COVID-19, Medicare 2020–2021 matched cohort.**

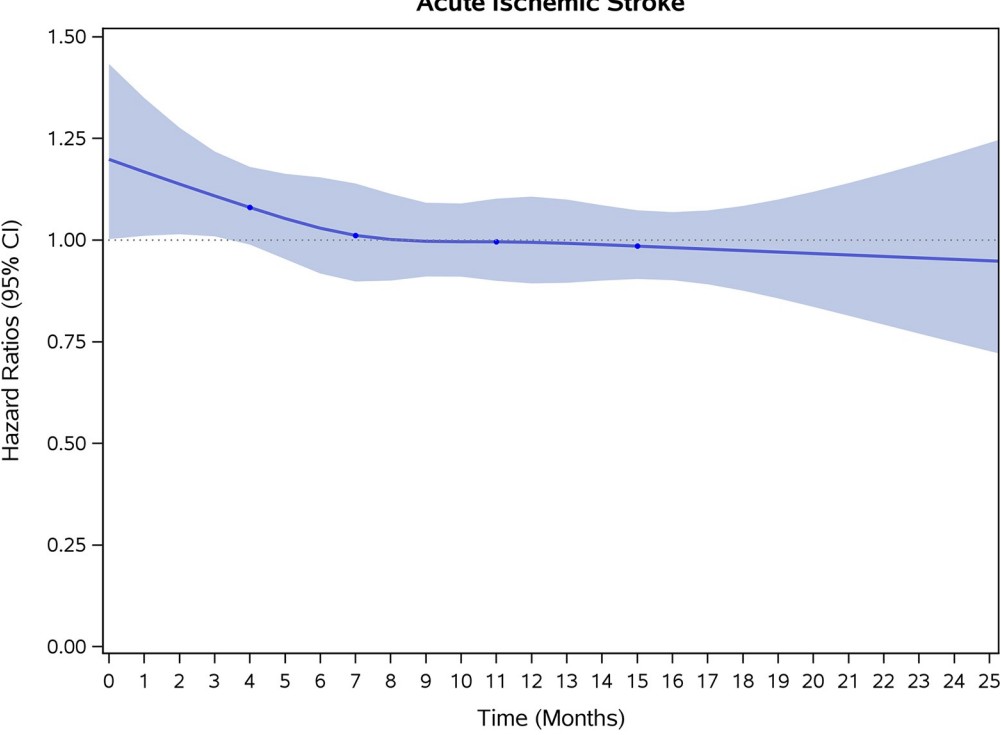

**Fig 15. Time-varying hazard ratios (95% CI) for risk of incident acute ischemic stroke associated with non-hospitalized COVID-19, Medicare 2020–2021 matched cohort.**

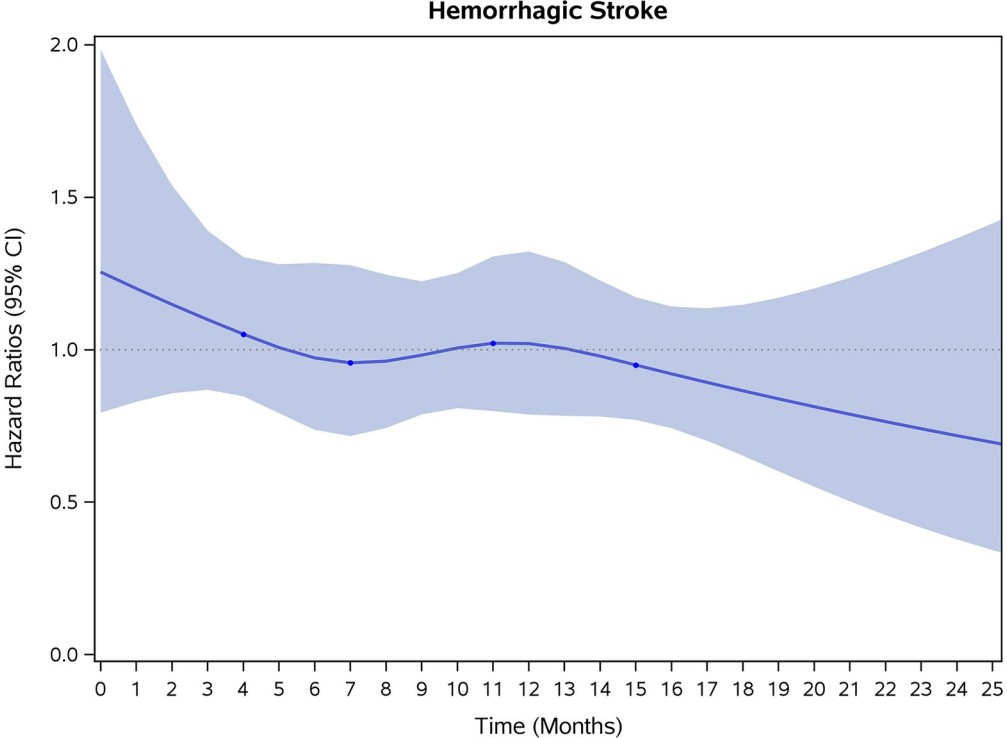

**Fig 16. Time-varying hazard ratios (95% CI) for risk of incident hemprrhagic stroke associated with non-hospitalized COVID-19, Medicare 2020–2021 matched cohort.**

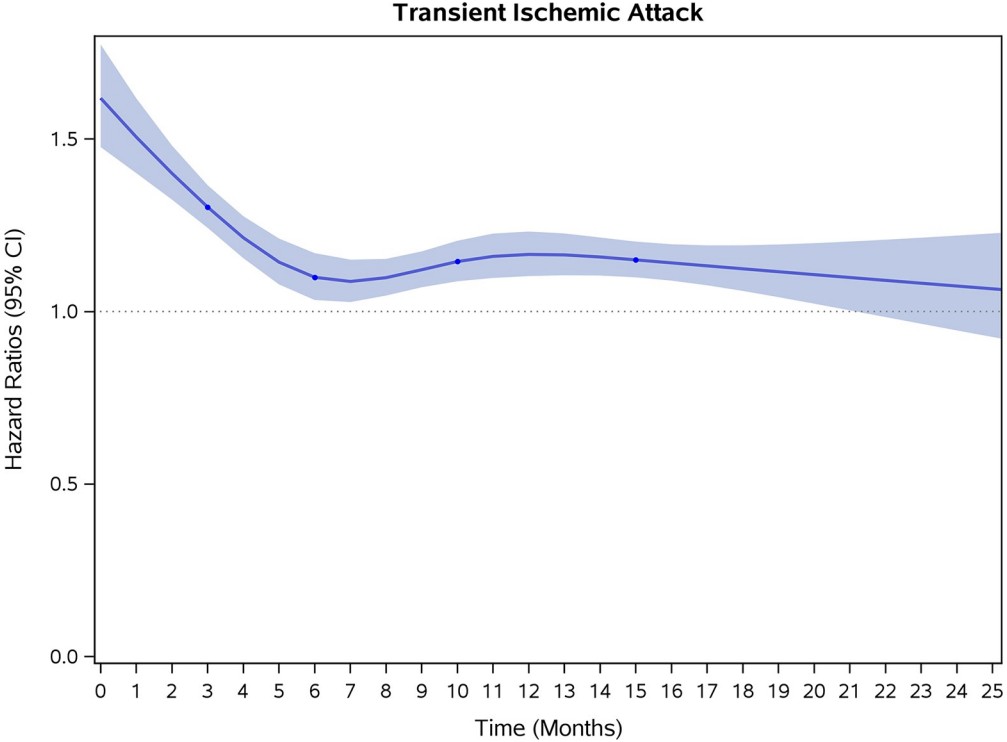

**Fig 17. Time-varying hazard ratios (95% CI) for risk of incident transient ischemic sttack associated with non-hospitalized COVID-19, Medicare 2020–2021 matched cohort.**

**Table 2. Incidence rates and adjusted hazard ratio (95% CI) for risk of death, hospitalization, CVD and stroke associated with non-hospitalized COVID-19 by follow-up time, Medicare 2020–2021 matched cohort.**

| Conditions [a] | 0–3 month | | | 4–6 month | | | ≥7 month | | |
|---|---|---|---|---|---|---|---|---|---|
| | COVID-19 status | | | COVID-19 status | | | COVID-19 status | | |
| | Yes | No | HR (95% CI) [b] | Yes | No | HR (95% CI) [b] | Yes | No | HR (95% CI) [b] |
| **Mortality** | | | | | | | | | |
| Events (person-years) [c] | 12,206 (234969) | 10,631 (235,202) | | 14,480 (231,845) | 14,949 (232,195) | | 55,878 (931,876) | 56,054 (932,451) | |
| Incidence rate per 1,000 person-years (95% CI) | 51.9 (51.0–52.9) | 45.2 (44.3–46.1) | 1.05 (1.01–1.09) | 62.5 (61.4–63.5) | 64.4 (63.4–65.4) | 0.88 (0.85–0.91) | 60.0 (59.5–60.5) | 60.1 (59.6–60.6) | 0.88 (0.87–0.90) |
| **Hospitalization** | | | | | | | | | |
| Events (person-years) [c] | 60,057 (227953) | 41,275 (231,054) | | 41,308 (217,491) | 36,994 (222,580) | | 120,309 (814,142) | 112,785 (845,190) | |
| Incidence rate per 1,000 person-years (95% CI) | 263.5 (261.4–265.6) | 178.6 (176.9–180.4) | 1.42 (1.40–1.45) | 189.9 (188.1–191.8) | 166.2 (164.5–167.9) | 1.11 (1.09–1.13) | 147.8 (146.9–148.6) | 133.4 (132.7–134.2) | 1.08 (1.07–1.09) |
| **Cardiovascular outcomes** | | | | | | | | | |
| **Abnormality of heart rhythm** | | | | | | | | | |
| Events (person-years) [c] | 247,84 (206,789) | 14,925 (211,875) | | 14,736 (200,544) | 12,573 (206,463) | | 36,180 (774,883) | 32,739 (802,406) | |
| Incidence rate per 1,000 person-years (95% CI) | 119.9 (118.4–121.4) | 70.4 (69.3–71.6) | 1.69 (1.64–1.74) | 73.5 (72.3–74.7) | 60.9 (59.8–62.0) | 1.20 (1.16–1.24) | 46.7 (46.2–47.2) | 40.8 (40.4–41.2) | 1.14 (1.12–1.16) |
| **AMI** | | | | | | | | | |
| Events (person-years) [c] | 1,948 (222,974) | 1,622 (224,084) | | 1,777 (219,922) | 1,685 (221,135) | | 6,722 (882,462) | 6,708 (886,647) | |
| Incidence rate per 1,000 person-years (95% CI) | 8.7 (8.4–9.1) | 7.2 (6.9–7.6) | 1.18 (1.07–1.29) | 8.1 (7.7–8.5) | 7.6 (7.3–8.0) | 1.04 (0.94–1.14) | 7.6 (7.4–7.8) | 7.6 (7.4–7.7) | 0.99 (0.94–1.03) |
| **AFIB and Flutter** | | | | | | | | | |
| Events (person-years) [c] | 9,616 (189754) | 5,725 (192,599) | | 5,785 (186,578) | 5,556 (189,669) | | 1,9626 (742,676) | 18,760 (755,169) | |
| Incidence rate per 1,000 person-years (95% CI) | 50.7 (49.7–51.7) | 29.7 (29.0–30.5) | 1.69 (1.61–1.76) | 31.0 (30.2–31.8) | 29.3 (28.5–30.1) | 1.05 (1.00–1.10) | 26.4 (26.1–26.8) | 24.8 (24.5–25.2) | 1.05 (1.03–1.08) |
| **Cardiac Arrhythmia** | | | | | | | | | |
| Events (person-years) [c] | 19,779 (202,321) | 12,018 (206,595) | | 13,496 (196,949) | 12,315 (201,752) | | 45,980 (761,151) | 43,517 (781,791) | |
| Incidence rate per 1,000 person-years (95% CI) | 97.8 (96.4–99.1) | 58.2 (57.1–59.2) | 1.67 (1.61–1.72) | 68.5 (67.4–69.7) | 61.0 (60.0–62.1) | 1.12 (1.08–1.16) | 60.4 (59.9–61.0) | 55.7 (55.1–56.2) | 1.08 (1.06–1.10) |
| **Cardiomyopathy** | | | | | | | | | |
| Events (person-years) [c] | 5,669 (225,045) | 4,532 (225,718) | | 3,720 (221,305) | 3,387 (222,156) | | 8,699 (884,124) | 8,681 (887,047) | |
| Incidence rate per 1,000 person-years (95% CI) | 25.2 (24.5–25.9) | 20.1 (19.5–20.7) | 1.23 (1.16–1.29) | 16.8 (16.3–17.4) | 15.2 (14.7–15.8) | 1.08 (1.01–1.15) | 9.8 (9.6–10.0) | 9.8 (9.6–10.0) | 0.98 (0.94–1.03) |
| **DVT** | | | | | | | | | |
| Events (person-years) [c] | 326 (234,569) | 215 (234,865) | | 295 (231,410) | 260 (231,837) | | 1016 (929,664) | 911 (930,669) | |
| Incidence rate per 1,000 person-years (95% CI) | 1.4 (1.2–1.5) | 0.9 (0.8–1.0) | 1.47 (1.15–1.87) | 1.3 (1.1–1.4) | 1.1 (1.0–1.3) | 1.10 (0.87–1.39) | 1.1 (1.0–1.2) | 1.0 (0.9–1.0) | 1.07 (0.95–1.22) |

(*Continued*)

**Table 2.** (Continued)

| Conditions [a] | 0–3 month COVID-19 status | | | 4–6 month COVID-19 status | | | ≥7 month COVID-19 status | | |
|---|---|---|---|---|---|---|---|---|---|
| | Yes | No | HR (95% CI) [b] | Yes | No | HR (95% CI) [b] | Yes | No | HR (95% CI) [b] |
| **PE** | | | | | | | | | |
| Events (person-years) [c] | 1,919 (233580) | 732 (234,175) | | 952 (230,294) | 805 (231,070) | | 3397 (924,060) | 3110 (926,559) | |
| Incidence rate per 1,000 person-years (95% CI) | 8.2 (7.9–8.6) | 3.1 (2.9–3.4) | 2.55 (2.26–2.87) | 4.1 (3.9–4.4) | 3.5 (3.3–3.7) | 1.15 (1.01–1.31) | 3.7 (3.6–3.8) | 3.4 (3.2–3.5) | 1.06 (0.99–1.14) |
| **HF** | | | | | | | | | |
| Events (person-years) [c] | 7,723 (174,171) | 4,818 (177,732) | | 5,176 (171,706) | 4,524 (175,571) | | 15,197 (685,180) | 14,997 (702,170) | |
| Incidence rate per 1,000 person-years (95% CI) | 44.3 (43.4–45.3) | 27.1 (26.4–27.9) | 1.63 (1.55–1.71) | 30.1 (29.3–31.0) | 25.8 (25.0–26.5) | 1.17 (1.10–1.23) | 22.2 (21.8–22.5) | 21.4 (21.0–21.7) | 1.04 (1.01–1.07) |
| **Hypercoagulability** | | | | | | | | | |
| Events (person-years) [c] | 3,495 (231,502) | 1,879 (232,473) | | 2,500 (228,084) | 2,220 (229,246) | | 9,442 (912,582) | 8,259 (917,311) | |
| Incidence rate per 1,000 person-years (95% CI) | 15.1 (14.6–15.6) | 8.1 (7.7–8.5) | 1.80 (1.66–1.94) | 11.0 (10.5–11.4) | 9.7 (9.3–10.1) | 1.09 (1.00–1.18) | 10.3 (10.1–10.6) | 9.0 (8.8–9.2) | 1.10 (1.06–1.15) |
| **IHD** | | | | | | | | | |
| Events (person-years) [c] | 8,163 (119,546) | 4,693 (122,067) | | 5,676 (117,417) | 4,385 (120,364) | | 16,373 (462,193) | 14,355 (476,887) | |
| Incidence rate per 1,000 person-years (95% CI) | 68.3 (66.8–69.8) | 38.4 (37.4–39.6) | 1.77 (1.68–1.86) | 48.3 (47.1–49.6) | 36.4 (35.4–37.5) | 1.32 (1.25–1.40) | 35.4 (34.9–36.0) | 30.1 (29.6–30.6) | 1.17 (1.14–1.21) |
| **PVD** | | | | | | | | | |
| Events (person-years) [c] | 12,729 (170,863) | 8,364 (174,589) | | 8,736 (167,395) | 6,912 (171,795) | | 20,010 (657,457) | 17,399 (679,622) | |
| Incidence rate per 1,000 person-years (95% CI) | 74.5 (73.2–75.8) | 47.9 (46.9–48.9) | 1.56 (1.50–1.62) | 52.2 (51.1–53.3) | 40.2 (39.3–41.2) | 1.30 (1.25–1.36) | 30.4 (30.0–30.9) | 25.6 (25.2–26.0) | 1.20 (1.16–1.23) |
| **Cerebrovascular outcomes** | | | | | | | | | |
| **All Stroke** | | | | | | | | | |
| Events (person-years) [c] | 1,144 (188,394) | 1,152 (191,102) | | 1,237 (186,372) | 1,230 (189,116) | | 4,731 (749,164) | 4,777 (759,781) | |
| Incidence rate per 1,000 person-years (95% CI) | 6.1 (5.7–6.4) | 6.0 (5.7–6.4) | 1.00 (0.89–1.12) | 6.6 (6.3–7.0) | 6.5 (6.2–6.9) | 1.01 (0.91–1.13) | 6.3 (6.1–6.5) | 6.3 (6.1–6.5) | 1.00 (0.94–1.06) |
| **AIS** | | | | | | | | | |
| Events (person-years) [c] | 812 (188,428) | 855 (191,131) | | 835 (186,470) | 829 (189,203) | | 3142 (750,456) | 3196 (7609,51) | |
| Incidence rate per 1,000 person-years (95% CI) | 4.3 (4.0–4.6) | 4.5 (4.2–4.8) | 0.95 (0.83–1.09) | 4.5 (4.2–4.8) | 4.4 (4.1–4.7) | 1.01 (0.88–1.16) | 4.2 (4.0–4.3) | 4.2 (4.1–4.3) | 0.99 (0.92–1.06) |
| **Hemorrhagic Stroke** | | | | | | | | | |
| Events (person-years) [c] | 130 (188,510) | 119 (191,210) | | 129 (186,672) | 160 (189,408) | | 539 (752,742) | 551 (763,268) | |
| Incidence rate per 1,000 person-years (95% CI) | 0.7 (0.6–0.8) | 0.6 (0.5–0.7) | 1.09 (0.77–1.55) | 0.7 (0.6–0.8) | 0.8 (0.7–1.0) | 0.81 (0.58–1.12) | 0.7 (0.7–0.8) | 0.7 (0.7–0.8) | 0.98 (0.83–1.16) |
| **TIA** | | | | | | | | | |

(*Continued*)

**Table 2.** (Continued)

| Conditions [a] | 0–3 month | | | 4–6 month | | | ≥7 month | | |
|---|---|---|---|---|---|---|---|---|---|
| | COVID-19 status | | | COVID-19 status | | | COVID-19 status | | |
| | Yes | No | HR (95% CI) [b] | Yes | No | HR (95% CI) [b] | Yes | No | HR (95% CI) [b] |
| Events (person-years) [c] | 3,935 (229,167) | 2,689 (230,219) | | 3,230 (225,502) | 2,812 (226,772) | | 11,628 (897,866) | 10,018 (903,627) | |
| Incidence rate per 1,000 person-years (95% CI) | 17.2 (16.6–17.7) | 11.7 (11.2–12.1) | 1.45 (1.36–1.56) | 14.3 (13.8–14.8) | 12.4 (12.0–12.9) | 1.15 (1.07–1.23) | 13.0 (12.7–13.2) | 11.1 (10.9–11.3) | 1.16 (1.12–1.21) |

Abbreviations: AFIB, Atrial Fibrillation; AIS, Acute Ischemic Stroke; AMI, Acute Myocardial Infarction; CI, confidence interval; DVT, Deep Vein Thrombosis; HF, Heart Failure; HR, hazard ratio; IHD, Ischemic Heart Disease; PE, Pulmonary Embolism; PVD, Peripheral Vascular Disease; TIA, Transient Ischemic Attack.

[a] For mortality and hospitalization, the analyses included full cohort of FFS beneficiaries with non-hospitalized COVID-19 and matched controls; for incidence of CVD and stroke, the analyses excluded FFS beneficiaries who had the CVD or stroke at baseline and cohort size varied by CVD and stroke outcomes.

[b] In addition to propensity score matching, HRs were adjusted for all matching variables to control for potential residual confounding. 95% CIs were Bonferroni corrected 95% CI.

[c] Incident events and person-years were calculated separately for 0–3, 4–6 and ≥7-month follow-up.

nonsignificant for mortality, AMI, cardiomyopathy, DVT and PE after 6 months from the COVID-19 index date.

Risk of mortality appeared to be lower among FFS beneficiaries with non-hospitalized COVID-19 compared to the matched controls after 3 months of follow-up. The reasons for the lowered mortality are not clear. One possible explanation may be that among all FFS beneficiaries who had COVID-19, the sicker patients tended to be hospitalized and the non-hospitalized patients may represent a healthier subset compared to all FFS beneficiaries. The present study showed nonsignificant association between COVID-19 and risk of all-stroke, AIS, and hemorrhagic stroke. Other studies showed significant time-varying association between COVID-19 and stroke, with the highest risk of stroke in first week after COVID-19 then reduced afterwards [14, 15]. In our study, we excluded the beneficiaries who had inpatient claims (where the incident stroke were recorded) within two weeks from the index date (to exclude possible hospitalized COVID-19 patients) which may partly explain the nonsignificant association.

For the stratified analyses of incident CVD and stroke by age group, sex and race/ethnicity, we observed several significant interactions between COVID-19 and subgroups. For example, the risk of mortality and PVD appeared to be higher among the beneficiaries aged ≥85 years; risk of abnormality of heart rhythms, AFIB, cardiomyopathy, TIA and IHD appeared to be higher among beneficiaries aged <85 years, and men had higher risk of mortality, hospitalizations, AMI, HF but lower PVD compared to women. The US Veterans cohort study of long-term cardiovascular outcomes of COVID-19 showed largely consistent risk of CVD and stroke associated with COVID-19 by age group, sex and race/ethnicity [8]. Other studies of long-term effects of COVID-19 on health outcomes with limited CVD and stroke outcomes also suggested the nonsignificant differences in risk by subgroups [9, 11]. Further studies are needed to clarify the potential differential risk of CVD and stroke associated with COVD-19 by subgroups.

Many studies have provided evidence of association between systemic infection—such as influenza, systemic respiratory tract infection, acute bronchitis, and herpes zoster—and increased risk of myocardial infarction and stroke [21–24]. Several mechanisms are proposed to link infections and increased risk of AMI and stroke including infection-induced systemic inflammation, atherosclerotic plaque instability, and increased risk of plaque rupture,

especially in the first few weeks after infections [25]. The pattern of initial stronger effects of COVID-19 on AMI in our study was consistent with other infections such as influenza [24, 25]. A recent study showed that SARS-CoV-2 directly infects the coronary vessels and induces atherosclerotic plaque inflammation. These changes could trigger acute cardiovascular complications and increase the risk of CVD [26]. The direct effect of COVID-19 on coronary vessels and plaque inflammation, especially among those who had cardiovascular risk factors such as hypertension, diabetes, older age, or with preexisting CVD or stroke [2, 4, 27], may help to explain the initial stronger effects of COVID-19 on CVD observed in the present study. However, COVID-19 can affect multiple organ systems and has numerous persistent symptoms, including long-term complications and increased risk of CVD and stroke (beyond 4 weeks from the onset of symptoms) [2, 4]. There are several other proposed mechanisms by which COVID-19 could be associated with increased risk of CVD and stroke. COVID-19 may have indirect effects through systemic inflammatory responses to the infection or cytokine storm that could lead to cell death and multiorgan dysfunctions or through the hypoxia-induced myocardial injury due to hypoxic respiratory failure and hypoxemia, or small vessel ischemia due to microvascular injury and thrombosis [2, 4, 27–29]. Further studies are needed to understand the underlying mechanisms of acute and long COVID-19 and its cardiovascular sequelae.

CVD is the leading cause of death in the United States with more than 934,000 deaths in 2021 [13, 30]. Stroke is the fifth leading cause of death and leading cause of serious long-term disability with approximately 795,000 stroke occurrences and more than 162,000 deaths in 2021 [13, 30]. COVID-19 has long-term effects on risk of CVD and stroke among hospitalized and non-hospitalized patients [1, 3, 5, 6, 8–12, 14, 15]. Although the overwhelming majority of people with COVID-19 have mild illness and do not require hospitalizations [16], it is important to recognize the increased risk of CVD among surviving non-hospitalized patients with COVID-19, especially among adults aged ≥65 years. Studies suggested that COVID-19 may affect more of those who had cardiovascular risk factors or with preexisting CVD or stroke [27]. Major modifiable risk factors for CVD and stroke are high blood pressure, high low-density lipoprotein (LDL) cholesterol, diabetes, obesity, smoking, poor sleep, unhealthy diet, and physical inactivity [31]. The majority of CVD and stroke occurrences may be attributed to these risk factors [13]. However, studies have suggested that high blood pressure treatment and control, as well as factors such as mental health, physical activity, diet, and sleep quality, were adversely impacted by the COVID-19 pandemic; moreover, these risk factors may have disproportionately affected people of color [32–34]. Other studies documented the significant racial disparities in CVD and stroke mortality during the COVID-19 pandemic [35, 36]. It is critically important to implement effective strategies for prevention, control, and treatment of major CVD risk factors amid the continuing increased number of individuals infected with SARS-CoV-2.

## Strengths and limitations

Our study included a large, population-based cohort of Medicare FFS beneficiaries aged ≥66 years with up to 26 months of follow-up. We used propensity score to match the comparable controls with a comprehensive list of matching variables to account for confounding and selection bias.

This study has several limitations. First, we used Medicare real-time GV preliminary data that is updated on monthly basis. Mortality and outcomes data may be subject to change when the datasets are finalized. However, CMS indicated that >98% of Medicare FFS claims were received within 6 months [37]. Second, Medicare beneficiaries with COVID-19 were identified through administrative claims and may be subject to misclassification. However, studies

suggested that physicians and hospitals were likely to follow the recommendations and guidelines regarding COVID-19 diagnosis because of the seriousness of the COVID-19 pandemic [38]. Third, testing for COVID-19 in the outpatient settings was limited and turnaround time for test results varied during the early phase of the pandemic that may have affected the accurate timing of COVID-19 exposure. Fourth, as there has not been clarity or consensus on definitions of cardiovascular conditions related to COVID-19 [3], we used CCW Condition Algorithms to define CVD and stroke, and used the published literature to define cardiovascular conditions that were not included in CCW Condition Algorithms. The definitions of COVID-19 related CVD and stroke may be subject to change. Fifth, our cohort included the early phase COVID-19 patients (April 1, 2020, to April 30, 2021) before the the Delta variant predominance that started in July 15, 2021, and findings may not apply to other variants of COVID-19. In addition, there is no detailed information of COVID-19 variants in Medicare data for the early phase of COVID-19, we were unable to stratify the analysis by strain of COVID-19. Sixth, the overwhelming majority of Medicare FFS beneficiaries with non-hospitalized COVID-19 in our cohort were unvaccinated (vaccination started in December 2020) and as COVID-19 vaccination information was incomplete in Medicare claims, we could not examine the effect of COVID-19 vaccinations on risk of CVD and stroke [39]. Seventh, Medicare FFS beneficiaries may not represent the general population.

Our results suggest that COVID-19 had time-varying effects on mortality, hospitalization, and incidence of CVD among Medicare FFS beneficiaries with non-hospitalized COVID-19. Patients and clinicians should understand the increased risk of CVD following COVID-19, especially amid the increasing numbers of COVID-19 patients with mild symptoms without requiring hospitlization. Our findings may inform efforts to care for patients with COVID-19 and highlight the potential importance of diagnosing and managing CVD risk factors among those patients.

## Supporting information

**S1 Fig. Cause-specific time-varying hazard ratios (95% CI) for risk of death, hospitalization, incident CVD and stroke associated with non-hospitalized COVID-19 by age group, Medicare 2020–2021 matched cohort.**
(DOCX)

**S2 Fig. Cause-specific time-varying hazard ratios (95% CI) for risk of death, hospitalization, incident CVD and stroke associated with non-hospitalized COVID-19 by sex, Medicare 2020–2021 matched cohort.**
(DOCX)

**S3 Fig. Cause-specific time-varying hazard ratios (95% CI) for risk of death, hospitalization, incident CVD and stroke associated with non-hospitalized COVID-19 by race/ethnicity, Medicare 2020–2021 matched cohort.**
(DOCX)

**S4 Fig. Cause-specific time-varying hazard ratios (95% CI) for risk of CVD and stroke associated with non-hospitalized COVID-19 including FFS Beneficiaries with pre-existing CVD and stroke, Medicare 2020–2021 matched cohort.**
(DOCX)

**S1 Table. Covariates used in propensity score matching, Medicare 2020–2021 matched cohort.**
(DOCX)

**S2 Table. ICD-10, CPT4 and HCPCS codes of chronic and other conditions based on CMS chronic condition warehouse condition algorithms (February 2021) and published literatures.**
(DOCX)

**S3 Table. Characteristics of original cohort Medicare FFS beneficiaries aged 66 years or older compared to non-hospitalized COVID-19 beneficiaries and matched controls, Medicare 2020–2021 matched cohort.**
(DOCX)

**S4 Table. Incidence rates of any diagnosis and adjusted hazard ratio (95% CI) for risk of CVD and stroke associated with non-hospitalized COVID-19 by follow-up time (including FFS beneficiaries with pre-existing CVD and stroke), Medicare 2020–2021 matched cohort.**
(DOCX)

## Acknowledgments

We thank Linda Schieb, Division for Heart Disease and Stroke Prevention, Centers for Disease Control and Prevention, for providing information on US family income by ZIP code of residences.

**Disclaimer:** The findings and conclusions in this article are those of the authors and do not necessarily represent the official position of the US Centers for Disease Control and Prevention (CDC).

## Author Contributions

**Conceptualization:** Quanhe Yang.

**Formal analysis:** Quanhe Yang, Anping Chang, Xin Tong.

**Funding acquisition:** Robert K. Merritt.

**Methodology:** Quanhe Yang, Sandra L. Jackson.

**Software:** Anping Chang.

**Supervision:** Quanhe Yang, Robert K. Merritt.

**Validation:** Xin Tong.

**Writing – original draft:** Quanhe Yang.

**Writing – review & editing:** Xin Tong, Sandra L. Jackson, Robert K. Merritt.

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
