## [Decision Letter · Decision Letter 0]

18 Mar 2024

PONE-D-23-36042Long-Term Cardiovascular Disease Outcomes in Non-Hospitalized Medicare Beneficiaries Diagnosed with COVID-19: Population-Based Matched Cohort StudyPLOS ONE

Dear Dr. Yang,

Thank you for submitting your manuscript to PLOS ONE. After careful consideration, we feel that it has merit but does not fully meet PLOS ONE’s publication criteria as it currently stands. Therefore, we invite you to submit a revised version of the manuscript that addresses the points raised during the review process.

We look forward to receiving your revised manuscript.

Kind regards,

Daniel Antwi-Amoabeng, MD, MSc

Academic Editor

PLOS ONE

Journal Requirements:

Additional Editor Comments (if provided):

Academic Editor’s Comments:

I commend the authors on the application of rigorous standard statistical analysis.

This is a noteworthy subject, which was dealt with in a structured and skillful manner.

Here are some suggestions:

Please provide a definition of this ICD-10 code on first use: “B97.29”

161: Please provide the URL as a reference or place it in the supplementary file.

321: Please delete the URL in the parenthesis.

359: Please delete the URL in the parenthesis and a provide proper citation.

Please discuss the probable mechanisms underlying the observed time-varying effect of SARS-CoV-2 infection on the CVD outcomes.

Reviewers' comments:

Reviewer's Responses to Questions

**Comments to the Author**

1. Is the manuscript technically sound, and do the data support the conclusions?

Reviewer #1: Yes

Reviewer #2: Yes

2. Has the statistical analysis been performed appropriately and rigorously? 

Reviewer #1: Yes

Reviewer #2: Yes

3. Have the authors made all data underlying the findings in their manuscript fully available?

Reviewer #1: Yes

Reviewer #2: Yes

4. Is the manuscript presented in an intelligible fashion and written in standard English?

Reviewer #1: Yes

Reviewer #2: Yes

5. Review Comments to the Author

Reviewer #1: Good timely and needy work for the current phase.

Do all the confounding factors have been analysed and excluded.

Is there any strain specificity with the CV outcomes of the patients in long run follow up.

Reviewer #2: Findings of this study is of great value and fulfill gaps existing in literature. Statistical analysis is rigorous and support presented data. I dont have any additional comments or suggestions to improve the text of the articles. Great job!

6. PLOS authors have the option to publish the peer review history of their article (what does this mean?). If published, this will include your full peer review and any attached files.

Reviewer #1: No

Reviewer #2: No

---

## [Author Response · Author response to Decision Letter 0]

3 Apr 2024

PONE-D-23-36042: Long-Term Cardiovascular Disease Outcomes in Non-Hospitalized Medicare Beneficiaries Diagnosed with COVID-19: Population-Based Matched Cohort Study 

Response to Editor’s comments

1. Please ensure that your manuscript meets PLOS ONE's style requirements, including those for file naming. The PLOS ONE style templates can be found at…

Response: we have followed the PLOS ONE’s style requirements in the revised version. 

Response: The Medicare data we have access to is the de-identified data, we can’t share these data because of the Data Use Agreement with the Centers for Medicare and Medicaid Services (CMS). Investigators can submit an application to CMS to access the Medicare data.

Response: At the end of Methods section, we revised the data availability statement as follows: 

Data Availability: The Medicare beneficiaries’ data used in this study to generate the findings are not publicly available and the authors cannot share these data due to the Data Use Agreement with the Centers for Medicare and Medicaid Services (CMS). Medicare data are available from CMS following a data use request from the third-party vendor, the Research Data Assistance Center (ResDAC). For instructions, guidance, and costs of CMS data, contact the ResDAC (http://www.resdac.org). 

Response: As described above, we can’t share the Medicare beneficiaries’ data because of the Data Use Agreement with CMS. We have updated our Data Availability statement.

Response: We have included the tables and figure captions for supporting information files and we have updated the in-text citations accordingly in the revised version.

Response: We have checked the references and reformatted the citations to PLOS ONE style. Please let me know if we may need to make any additional changes in the reference style.

Academic Editor’s Comments:

I commend the authors on the application of rigorous standard statistical analysis.

This is a noteworthy subject, which was dealt with in a structured and skillful manner.

Response: Thank you for your time to review the manuscript and providing thoughtful and constructive comments and suggestions.

1. Please provide a definition of this ICD-10 code on first use: “B97.29”

Response: We have added the definition for ICD-10 B97.29: “Other coronavirus as the cause of diseases classified elsewhere”

2. 161: Please provide the URL as a reference or place it in the supplementary file.

Response: We have changed the URL as a reference in the revised manuscript.

3. 321: Please delete the URL in the parenthesis.

Response: We have deleted this URL in the revised version.

4. 359: Please delete the URL in the parenthesis and a provide proper citation.

Response: We have deleted this URL and provided a citation.

5. Please discuss the probable mechanisms underlying the observed time-varying effect of SARS-CoV-2 infection on the CVD outcomes.

Response: Thank you for this thoughtful comment. Few COVID-19 and CVD studies have examined the time-varying effects of COVID-19 on risk of CVD and stroke. The probable mechanisms underlying the observed time-varying effects of SARS-CoV-2 infection on the CVD outcomes are unclear. In the Discussion, we have added: “A recent study showed that SARS-CoV-2 directly infects the coronary vessels and induces atherosclerotic plaque inflammation. These changes could trigger acute cardiovascular complications and increase the risk of CVD.[26] The direct effect of COVID-19 on coronary vessels and plaque inflammation, especially among those who had cardiovascular risk factors such as hypertension, diabetes, older age, or with preexisting CVD or stroke,[2, 4, 27] may help to explain the initial stronger effects of COVID-19 on CVD observed in the present study. However, COVID-19 can affect multiple organ systems and has numerous persistent symptoms, including long-term complications and increased risk of CVD and stroke (beyond 4 weeks from the onset of symptoms).[2, 4] There are several other proposed mechanisms by which COVID-19 could be associated with increased risk of CVD and stroke. COVID-19 may have indirect effects through systemic inflammatory responses to the infection or cytokine storm that could lead to cell death and multiorgan dysfunctions or through the hypoxia-induced myocardial injury due to hypoxic respiratory failure and hypoxemia, or small vessel ischemia due to microvascular injury and thrombosis.[2, 4, 27-29] Further studies are needed to understand the underlying mechanisms of acute and long COVID-19 and its cardiovascular sequelae.”

Reviewer #1: Good timely and needy work for the current phase.

Response: Thank you for reviewing the manuscript.

1. Do all the confounding factors have been analyzed and excluded.

Response: We have included a comprehensive list of demographic, socioeconomic and comorbidity conditions in the propensity score matching. The matching variables and potential confounding factors were from seven domains: Demographic factors; Socio-economic conditions; Healthcare utilization characteristics; Frailty characteristics; 37 Chronic Condition Warehouse (CCW) Conditions; CMS Hierarchical Condition Category (CMS-HCC) risk score; and Charlson Comorbidity Index. We have checked each variable before matching and excluded the incomplete potential confounders (excluded confounders are not shown in S1 Table). 

2. Is there any strain specificity with the CV outcomes of the patients in long run follow up.

Response: Thank you for the thoughtful comments. Our baseline cohort included the early phase COVID-19 patients from April 1, 2020 to April 30, 2021 before the Delta variant predominance that started in July 15, 2021. For early phase of COVID-19, there is no detailed information of COVID-19 variants in Medicare data. We have added a sentence to the limitations indicating that we were unable to stratify our analyses by strain of COVID-19.

Reviewer #2: Findings of this study is of great value and fulfill gaps existing in literature. Statistical analysis is rigorous and support presented data. I don’t have any additional comments or suggestions to improve the text of the articles. Great job!

Response: We appreciate that the reviewer sees the value of our study, and we are grateful for your review of our manuscript.

Reviewer #3: it’s a great pleasure to read this new area of interest topic.

Response: Thank you for your time to review the manuscript. 

1. Is there any strain specifically noted for the CV morbidity and mortality? 

Response: Thank you for the thoughtful comments. Our baseline cohort included the early phase COVID-19 patients from April 1, 2020 to April 30, 2021 before the Delta variant predominance that started in July 15, 2021. For early phase of COVID-19, there is no detailed information of COVID-19 variants in Medicare data. We have added a sentence to the limitations indicating that we were unable to stratify our analyses by strain of COVID-19.

2. Is there any data regarding worsening of previous CV patients?

Response: In the sensitivity analysis, we compared the incidence rates and adjusted hazard ratio for risk of CVD and stroke associated with COVID-19 by follow-up time by excluding pre-existing CVD and stroke vs. including pre-existing CVD and stroke; the pattern of association remained largely consistent. Here are a few selected conditions (in the table below), please see supporting information S4 Table and S2 Figure for more detailed results. 

Comparison of incidence rates and adjusted hazard ratio (95% CI) for risk of CVD and stroke associated with non-hospitalized COVID-19 by follow-up time: excluding pre-existing CVD and stroke vs. including pre-existing CVD and stroke), Medicare 2020–2021 Matched Cohort

---

## [Editor Report · Decision Letter 1]

9 Apr 2024

Long-Term Cardiovascular Disease Outcomes in Non-Hospitalized Medicare Beneficiaries Diagnosed with COVID-19: Population-Based Matched Cohort Study

PONE-D-23-36042R1

Dear Dr. Yang,

We’re pleased to inform you that your manuscript has been judged scientifically suitable for publication and will be formally accepted for publication once it meets all outstanding technical requirements.

Kind regards,

Daniel Antwi-Amoabeng, MD, MSc

Academic Editor

PLOS ONE
---

## [Editor Report · Acceptance letter]

26 Apr 2024

PONE-D-23-36042R1 

PLOS ONE

Dear Dr. Yang, 

I'm pleased to inform you that your manuscript has been deemed suitable for publication in PLOS ONE. Congratulations! Your manuscript is now being handed over to our production team.

Kind regards, 

on behalf of

Dr. Daniel Antwi-Amoabeng 

Academic Editor

PLOS ONE